# New Onset and Exacerbation of Autoimmune Bullous Dermatosis Following COVID-19 Vaccination: A Systematic Review

**DOI:** 10.3390/vaccines12050465

**Published:** 2024-04-26

**Authors:** Po-Chien Wu, I-Hsin Huang, Ching-Ya Wang, Ching-Chi Chi

**Affiliations:** 1Department of Dermatology, Chang Gung Memorial Hospital, Linkou Main Branch, Taoyuan 33305, Taiwan; pochienwu5@gmail.com (P.-C.W.); eugenia9797@gmail.com (I.-H.H.); 2Department of Dermatology, Heping Fuyou Branches, Taipei City Hospital Renai, Taipei 10629, Taiwan; rebbid1000@gmail.com; 3School of Medicine, College of Medicine, Chang Gung University, Taoyuan 33302, Taiwan

**Keywords:** autoimmune bullous dermatosis, bullous pemphigoid, mucous membrane pemphigoid, linear IgA bullous dermatosis, pemphigus vulgaris, pemphigus foliaceus, pemphigus erythematosus, pemphigus vegetans, COVID-19, vaccine

## Abstract

Background: Cases of autoimmune bullous dermatosis (AIBD) have been reported following COVID-19 vaccination. Objective: We aimed to provide an overview of clinical characteristics, treatments, and outcomes of AIBDs following COVID-19 vaccination. Methods: We conducted a systematic review and searched the Embase, Cochrane Library, and Medline databases from their inception to 27 March 2024. We included all studies reporting ≥ 1 patient who developed new-onset AIBD or experienced flare of AIBD following at least one dose of any COVID-19 vaccine. Results: We included 98 studies with 229 patients in the new-onset group and 216 in the flare group. Among the new-onset cases, bullous pemphigoid (BP) was the most frequently reported subtype. Notably, mRNA vaccines were commonly associated with the development of AIBD. Regarding the flare group, pemphigus was the most frequently reported subtype, with the mRNA vaccines being the predominant vaccine type. The onset of AIBD ranged from 1 to 123 days post-vaccination, with most patients displaying favorable outcomes and showing improvement or resolution from 1 week to 8 months after treatment initiation. Conclusions: Both new-onset AIBD and exacerbation of pre-existing AIBD may occur following COVID-19 vaccination. Healthcare practitioners should be alert, and post-vaccination monitoring may be essential.

## 1. Introduction

To mitigate the transmission of severe acute respiratory syndrome coronavirus 2 (SARS-CoV-2) [1,2,3], various vaccines have been rapidly developed, including mRNA vaccines (BioNTech/Pfizer (Comirnaty; BNT162b2) and Moderna (Spikevax; mRNA-1273)), viral-vectored vaccines (AstraZeneca (Covishield; AZD1222/ChAdOx1) and Johnson & Johnson (COVID-19 Vaccine Janssen; Ad26.COV2.S/JNJ-78436735)), and inactivated vaccines (Sinopharm (BBIBP-CorV) and Sinovac (CoronaVac)) [4,5,6]. With the introduction of global mass vaccination, reports of post-vaccination cutaneous adverse events have emerged, including injection site reactions, urticaria, and morbilliform eruptions [7,8,9,10,11]. Furthermore, cases of autoimmune bullous dermatosis (AIBD) have been documented [12,13,14,15].

AIBD is characterized by the presence of autoantibodies targeting specific adhesion molecules, such as desmoglein, BP180, or BP230, within the skin or mucosae [16]. Clinical manifestations of AIBD range from localized vesiculobullous eruption to widespread potentially life-threatening skin detachment [17]. Following COVID-19 vaccination, various subtypes of AIBD have been reported, including diseases with intraepidermal detachment, such as pemphigus vulgaris (PV), pemphigus foliaceus (PF), pemphigus erythematosus (PE), pemphigus vegetans (PVeg), as well as diseases with subepidermal detachment, such as bullous pemphigoid (BP), mucous membrane pemphigoid (MMP), and linear IgA bullous dermatosis (LABD) [6,18,19]. The potential association between COVID-19 vaccination and AIBD requires further investigation, and a comprehensive review of this topic is needed. Given the increasing number of COVID-19 vaccine administrations, we conducted a systematic review to provide an overview of the clinical characteristics, treatment, and outcomes of AIBDs following COVID-19 vaccination.

## 2. Methods

This systematic review was registered with PROSPERO (CRD42023390478), and it was performed in accordance with the updated Preferred Reporting Items for Systematic Reviews and Meta-Analyses (PRISMA) guidelines [20,21,22]. Comprehensive searches were performed in the Embase, Cochrane Library, and Medline databases from their inception to 27 March 2024 using relevant terms, including ‘COVID-19’, ‘vaccine’, ‘autoimmune bullous dermatosis’, ‘vesiculobullous skin diseases’, ‘pemphigus’, ‘pemphigus vulgaris’, ‘pemphigus foliaceus’, ‘pemphigus erythematosus’, ‘bullous pemphigoid’, ‘mucous membrane pemphigoid’, and ‘linear IgA bullous dermatosis’. These terms were applied as free text, medical subject headings (MeSH in PubMed and Emtree in Embase), and abbreviations in the literature search. Boolean operators were used to combine keywords, and a primary search strategy was developed without language or publication data limitations (Appendix A). Additionally, the reference lists of all identified articles were screened to identify further relevant studies.

We included studies reporting at least one patient who developed new-onset AIBD or experienced an exacerbation of AIBD following administration of at least one dose of any COVID-19 vaccine. Exacerbation was defined as the presence of increased body surface area involvement, the presence of vesiculobullous lesions or skin erythema, subjective worsening reported by the patient, worsening described in physical examination findings, or clinician assessment or plan indicating exacerbation, rebound, or worsening of AIBD compared to previous examination. Review articles, conference abstracts, and in vitro or animal model studies were excluded. Two experienced authors (Wu and Wang) independently conducted the literature search, data extraction, and quality assessments. Any discrepancies between the reviewers were resolved by a third author (Huang). The quality of case reports and series was assessed using the appraisal tool developed by Murad et al. [23], while observational studies were evaluated using the National Institute of Health quality assessment tool (Appendix A) [24]. 

Data extraction was performed independently by two authors (Wu and Wang) and included the following information from the included studies: author, year of publication, country, demographic information of patients (age and sex), blister sites, COVID-19 vaccination details (vaccine type and dose), onset time, classification of cases as new-onsets or exacerbations, AIBD subtype, other potential triggers, pathology examinations (Hematoxylin and Eosin stains and immunofluorescence study), enzyme-linked immunoassay (ELISA) results (such as BP180, BP 230, desmoglein [dsg] 1, and desmoglein 3), prior and post-exacerbation treatments, outcomes, and reactions to subsequent COVID-19 vaccination. The patient groups were further categorized based on the occurrence of new AIBD onset or exacerbation of AIBD, and all patients were classified according to AIBD subtypes.

## 3. Results

### 3.1. Literature Search

As shown in Figure 1, 333 studies were identified after searching three major databases and performing a manual search of the reference lists of identified studies. We excluded 91 studies as duplicates, and 75 studies were excluded for being unrelated to the study question after assessing the title or abstract. The full texts of the remaining 167 studies were reviewed, and 98 studies were identified as meeting the inclusion criteria for qualitative synthesis. A total of 74 studies reporting new-onset AIBD, 15 studies reporting exacerbation of AIBD, and 9 studies reporting both new onset and exacerbation of AIBD were included in this study (Table 1 and Table 2). The quality assessments of case reports and series consistently received scores ranging from five to seven according to the methodology proposed by Murad et al. [23]. For observational studies, all of the assessments were rated as ‘fair’ using the National Institute of Health quality assessment tool [24]. 

### 3.2. Patient Characteristics

Detailed patient information is presented in Table 1 and Table 2. The characteristics of the included studies are summarized in Table 3. The new-onset group comprised 229 patients, mostly from America, with ages ranging from 11 to 97 years. Although most studies did not report patients’ sex, a slight male predominance was noted among those that did. The most frequently encountered diagnosis in the group was BP in 174 patients, followed by PV in 23 and PF in 16.

The flare group included 216 patients, with ages ranging from 20 to 88 years, who primarily had pemphigus (specific subtype unspecified). Most patients were from Asia (44%) and America (41%). Similarly to the new-onset group, most studies did not provide information on patients’ sex, but a slight male predominance existed among those that did.

### 3.3. Vaccine Type, Vaccine Dose, and Time to AIBD Onset Following Vaccination

In the new-onset group, 55% of patients received the BioNTech/Pfizer vaccine, followed by the Moderna vaccine (16%) and the Oxford-AstraZeneca vaccine (13%). However, it is noteworthy that the vaccine type was not reported for a large number of patients. Most cases of new-onset AIBD occurred after the second (39%) or first vaccine dose (34%), while 15% of AIBD patients experienced onset following both doses. The onset times varied widely, ranging from 1 to 123 days after vaccination.

In the flare group, most patients were administered the BioNTech/Pfizer vaccine (56%), followed by the Sinovac vaccine (18%) and the Moderna vaccine (16%). Flares were most frequently reported after the third vaccine dose (63%), followed by the first dose (24%) and the second dose (10%). The onset of AIBD symptoms ranged from 1 day to 92 days following vaccination. 

### 3.4. Other Potential Non-Vaccine Triggers 

In the new-onset group, most studies did not provide information on other potential non-vaccine triggers. However, some BP patients had pre-existing neurological or psychiatric disorders, such as dementia, depression, or Alzheimer’s disease, which are known to be associated with the development of BP [28,114,115]. Additionally, dipeptidyl peptidase 4 (DPP-4) inhibitors, a well-established risk factor for BP [116], were used by some patients [34,44,45,48]. In the majority of cases, patients denied any new medication use. 

In the flare group, the information regarding other potential triggers was unavailable in most studies. Nevertheless, two patients had a history of COVID-19 infection prior to receiving the COVID-19 vaccines, and subsequently experienced a BP eruption [99,103]. 

### 3.5. The Assessment of Naranjo Scores for New-Onset AIBD or AIBD Flares

To evaluate the potential causal relationship between COVID-19 vaccination and AIBD development, we applied the Naranjo scores to all cases (Appendix A) [117]. In the new-onset group, 87% of cases were categorized as ‘possible’, and 13% as ‘probable’. In the flare group, 92% of cases were classified as ‘possible’, and 8% as ‘probable’. Notably, all cases deemed ‘probable’ in causality had experienced a disease flare following both doses of COVID-19 vaccines, contributing to the overall score for these cases [6,13,25,37,38,40,41,42,49,50,53,54,68,71,75,86,91].

### 3.6. Treatment and Outcomes for New-Onset AIBD or AIBD Flares

In the new onset group, BP patients with limited involvement were treated with topical corticosteroids, while those with more extensive involvements received a variety of systemic immunomodulators, including corticosteroids, doxycycline, nicotinamide, methotrexate, azathioprine, cyclosporine, mycophenolate mofetil, cyclophosphamide, dapsone, colchicine, or hydroxychloroquine [5,6,14,15,28,31,32,35,38,39,42,43,44,45,49,51,53,56,70,71,72,73,75,77,82,86,87,92,93]. DPP-4 inhibitors were suspended in patients using these medications [34,44,45,48]. Intravenous immunoglobulin G (IVIG) was administered in selected cases, and biologics, such as dupilumab and omalizumab, were utilized [26,34,49,55,56]. Rituximab was introduced in three cases, leading to significant improvement [51,56,63]. Most patients with pemphigus were managed with systemic corticosteroids and immunomodulators, with rituximab administered in 29% of cases [51,63,74,75,78,79,81,89]. In one case of PVeg, intralesional injections of onabotulinum toxin, corticosteroids, and mycophenolate mofetil were used, resulting in resolution after 6 months [87]. The majority of patients demonstrated improvement (56%) or resolution (35%) after treatment, with resolution times ranging from 1 week to 8 months. One case of BP showed improvement after prednisolone treatment, but the patient died due to pulmonary embolism one month after discharge [29]. Disease flare after both vaccine doses was observed in 69% of reported cases, but most studies lacked data on subsequent vaccinations.

In the flare group, the predominant treatment approach involved topical or systemic corticosteroids supplemented by immunomodulators, such as doxycycline, nicotinamide, methotrexate, azathioprine, or mycophenolate mofetil, in refractory cases [6,18,30,105,106]. Additional corticosteroid therapy was used in most patients experiencing a flare of AIBD, with further immunosuppressants utilized for treatment-resistant cases [14,18,32,106]. Rituximab was administered in six cases, resulting in four cases experiencing disease improvement; one case died 15 days after the administration of COVID-19 vaccination due to sepsis, and one case had ongoing treatment and no final outcome was reported [51,99,105,108,109,110]. The majority of cases showed improvement (65%) or resolution (22%) after treatment, with resolution times ranging from 1 to 10 weeks. Only three of the reported cases (20%) experienced a similar flare following their initial COVID-19 vaccination and exhibited disease exacerbation after the second dose [6,18,30,105].

## 4. Discussion

In this systematic review, we have compiled all available reports of new-onset AIBD or AIBD flares following COVID-19 vaccination. Our analysis included 98 studies, encompassing 229 patients in the new-onset group and 216 patients in the flare group. Among the new-onset cases, BP was the most frequently reported subtype, while pemphigus was the most commonly reported subtype in the flare group. As we know, clinical relapse is commonly seen in pemphigus, with a relapse rate as high as 82% [118]. The chronic and relapsing features of pemphigus may contribute to the larger number of flare cases relative to BP. Notably, both new onset and exacerbation of AIBDs were frequently observed following the administration of mRNA vaccines. However, we should recognize that mRNA vaccines were the most frequently administered vaccine worldwide. Onset time varied widely among both new-onset and flare groups, ranging from 1 to 123 days. Most patients achieved favorable outcomes, with improvement or resolution occurring within 1 week to 8 months after treatment initiation.

The potential association between vaccination and AIBD has been investigated in the previous research [119]. Various vaccines, including influenza, tetanus and diphtheria, hepatitis B, herpes zoster, and quadrivalent human papillomavirus, have been reported to be associated with AIBD development [120]. With the substantial increase in COVID-19 vaccinations, the link between newly developed vaccines and AIBD has been reexamined. The theory of molecular mimicry between specific basement membrane proteins and the spike protein of SARS-CoV-2 has been proposed as a potential cause [121]. Additionally, mRNA vaccines are suggested to activate pro-inflammatory pathways by interacting with toll-like receptors, potentially leading to increased production of interleukin (IL) -4, IL-17, interferon-γ, and tumor necrosis factor-α cytokines [71,79,122]. Because autoreactive T cells and the dysregulation of T helper (Th)1 and Th2 responses play a crucial role in both pemphigus and pemphigoid [123], the vaccine trigger and cytokine modulation may promote an imbalance between Th2 responses against cutaneous antigens, fostering the generation of autoreactive B cells and contributing to AIBD development [122]. Vaccine-induced inflammation may also disrupt the basement membrane, leading to the production of anti-basement membrane antibodies [121]. Furthermore, human leukocyte antigen (HLA) molecules, including alleles *HLA-DQB1*0503* and *HLA-DRB1*0402* in pemphigus, as well as *HLA-DQB1*0301* in pemphigoid, may represent key predisposing factors for drug-induced AIBDs [124]. However, none of the included cases underwent HLA examinations, necessitating further investigations. 

On the contrary, Birabaharan et al. conducted a cohort study involving over 1.5 million individuals who received mRNA COVID-19 vaccinations, which revealed no difference in the risk of new-onset BP within a 6-month period between vaccinated patients and those who remained unvaccinated [33]. Another investigation by Kasperkiewicz et al. demonstrated that circulating anti-SARS-CoV-2 antibodies did not cross-react with the main AIBD autoantigens, including dsg 1, dsg 3, envoplakin, BP180, BP230, and type VII collagen [125]. This perspective is consistent with the findings of previous systematic reviews, which posited that the hypothesized causal relationship is likely to be a relatively rare occurrence [126,127]. In our study, we not only included a substantially larger sample size compared to previous studies, but we also employed the Naranjo score to investigate causality. Patients with severe or extensive AIBD are usually advised against re-exposure to the same vaccine. However, in our study, 23 patients who experienced new onset or exacerbation of AIBD were re-exposed to the same vaccine, leading to recurrence. This implicates COVID-19 vaccines as the likely causative agents, supported by the high Naranjo rating score of 7. Our research provides evidence suggesting a potential association between COVID-19 vaccination and the development of AIBD to some extent, as indicated by the short onset interval and the absence of other triggers in most cases. These findings are in accordance with the previous literature, underscoring that mRNA vaccines were the most commonly reported vaccine type in both new onset and exacerbation of AIBD cases, followed by inactivated and viral-vectored vaccines [127]. 

It is worth noting that some studies reported potential non-vaccine triggers, such as neurological or psychiatric disorders, use of DPP-4 inhibitor, polypharmacy, or a history of COVID-19 infection [28,34,43,44,46,49,50,92,99]. The etiology and pathogenesis of AIBD remain largely elusive. However, the occurrence of exacerbation of AIBD has been reported in association with specific triggering factors, including medications, physical stimuli, infections, and organ transplantations [128]. We outlined these cases and assigned lower scores on the Naranjo score, which consequently decreased the overall rating. Only 13% of the new-onset AIBD patients and 8% of AIBD flare cases were rated as probable according to the Naranjo score. Nevertheless, it is essential to acknowledge that most studies did not report such triggers, limiting the calculation of the Naranjo score. Given that the existing data predominantly consist of anecdotal, single-case reports with a low level of evidence, real-world, population-based studies are warranted to elucidate a definitive link between COVID-19 vaccinations and risk of AIBD. However, this should not dissuade the current vaccination recommendations for patients with AIBD, given the favorable risk–benefit ratio. 

Our study has certain limitations. Firstly, most of the included studies were case reports, case series, and retrospective observational studies from database collections. Some studies lacked comprehensive documentation of patients’ clinical conditions, while others were deficient in critical information, including vaccine dosage, additional triggers, laboratory findings, treatment modalities, and disease outcomes. Secondly, not all studies presented results of skin biopsies, immunofluorescence studies, or ELISA tests, thereby raising questions about the accuracy of disease diagnoses in some cases. Thirdly, essential parameters for assessing disease severity in AIBD patients, such as the bullous pemphigoid disease area index (BPDAI), the pemphigus area and activity score (PAAS), and the percentage of body surface area affected, were not reported among all studies. These parameters are pivotal for evaluating disease severity before vaccination, after vaccination, and following treatment. Fourthly, only a limited number of cases provided information regarding whether patients received subsequent vaccine doses, and the duration of follow-up was relatively short. In our analysis, most patients in the new-onset and flare groups showed improvement or resolution. However, given the chronic and relapsing nature of AIBD, future long-term follow-up studies are imperative to establish a stronger evidence base, and ongoing monitoring is essential for these patients [129]. 

## 5. Conclusions

In conclusion, both new-onset AIBD and exacerbation of pre-existing AIBD may occur following COVID-19 vaccination. Healthcare practitioners should raise concerns for AIBD when administering COVID-19 vaccines, and post-vaccination monitoring may be essential. Current evidence continues to favor COVID-19 vaccination in individuals with AIBD, owing to its significant protective benefits against SARS-CoV-2. More studies are imperative to elucidate the underlying mechanisms of the association between COVID-19 vaccines and the development of AIBD.

## Figures and Tables

**Figure 1 vaccines-12-00465-f001:**
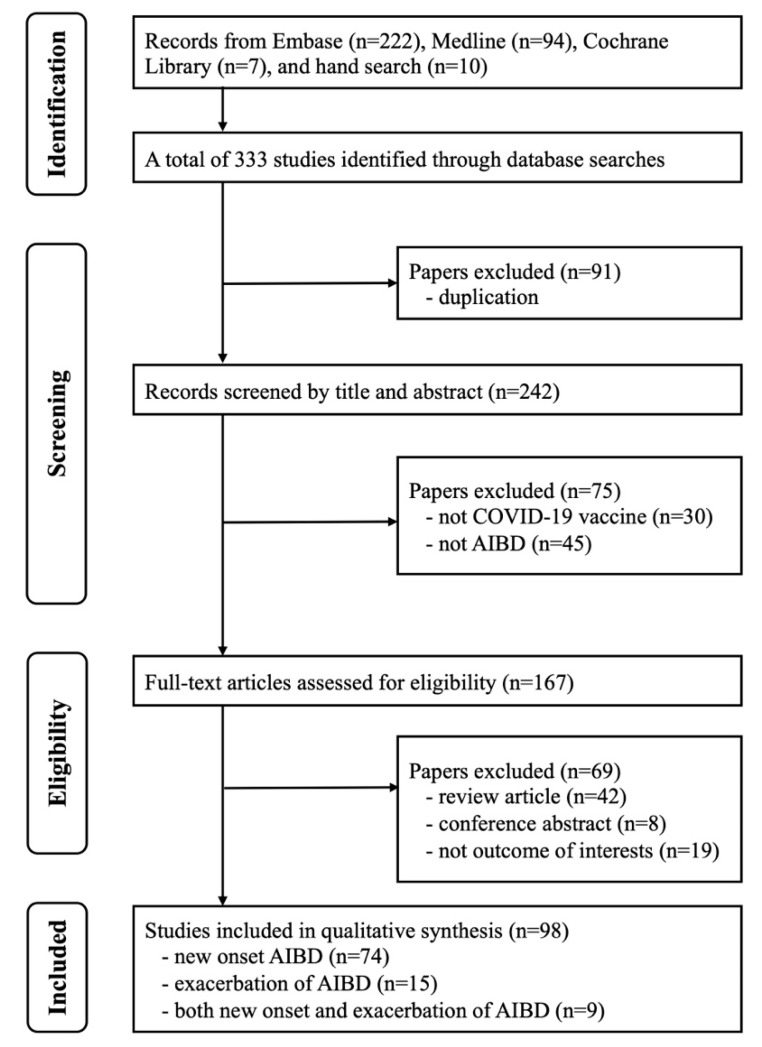
PRISMA flowchart of the selection of studies.

**Table 1 vaccines-12-00465-t001:** Characteristics of the included studies reporting new onset of autoimmune bullous dermatosis.

Author, Year	Country	Age, Sex	Blister Sites	Vaccine (Dose)	Onset	Other Triggers	Pathology	DIF/IIF	ELISA	Treatment	Outcome (Time)	Further Vaccine
BP												
Khalid 2021 [25]	US	62 M	1st: trunk 2nd: trunk, limbs, genitalia	MOD (both)	1st: 14 d 2nd: 4 d	No new/change in meds, allergic hx	eos	NR	NR	NR	NR	Flare after both doses
Nakamura 2021 [26]	Japan	83 F	Trunk and limbs	BNT (2nd)	3 d	No DPP4i use	SubE, eos	DIF: IgG (linear) IIF: NR	BP180+	SC, IVIG	Improved (NR)	NR
Pérez-López 2021 [27]	Spain	78 F	Face, trunk, and limbs	BNT (both)	1st: 3 d 2nd: NR	NR	NR	NR	NR	TC, SC	Improved (NR)	Flare after both doses
Tomayko 2021 [28]	US	97 F	NR	BNT (2nd)	2 d	NR	SubE, eos	DIF: C3/IgG/IgA (linear) IIF: NR	BP180+/230+	TC, DOX, NAM	Improved (2 w)	NR
	US	75 M	NR	BNT (2nd)	10 d	NR	SubE, eos	DIF: C3 (linear) IIF: NR	BP180+	TC, SC, DOX, NAM	Improved (3 w)	NR
	US	64 M	NR	BNT (2nd)	14 d	NR	SubE, eos	DIF: C3 (linear) IIF: NR	BP180+/230+	TC	Improved (4 w)	NR
	US	82 M	NR	BNT (2nd)	1 d	NR	SubE, eos	DIF: C3/IgG/IgA (linear) IIF: NR	BP180−/230−	TC	Resolved (2 w)	NR
	US	95 F	NR	BNT (1st)	5 d	NR	SubE, eos	DIF: C3/IgG/IgA (linear) IIF: NR	BP180−/230−	TC, DOX, NAM	Resolved (8 w)	No flare
	US	87 M	NR	MOD (2nd)	21 d	Alzheimer’s disease	SubE, eos	DIF: C3 (linear) IIF: NR	BP180+/230+	SC, DOX, NAM	Ongoing (105 d)	NR
	US	42 F	NR	MOD (2nd)	3 d	NR	SubE, eos	DIF: C3/IgG/IgM (granular) IIF: NR	BP180+/230+	TC, SC	Ongoing (23 d)	NR
	US	85 M	NR	BNT (1st)	5 d	NR	SubE, eos	DIF: C3/IgG (linear) IIF: NR	NR	SC	Ongoing (59 d)	Not received
	US	83 F	NR	MOD (1st)	8 d	Major depression	SubE, eos	DIF: Negative IIF: Negative	BP180−/230−	TC, SC	Ongoing (2 m)	Not received
	US	66 F	NR	BNT (both)	1st: 7 d 2nd: NR	NR	SubE, eos	DIF: Negative IIF: Negative	BP180−/230−	TC, SC	Resolved (3 w)	Flare after both doses
	US	70 F	NR	MOD (1st)	9 d	NR	SubE, eos	DIF: Negative IIF: NR	NR	SC	Resolved (15 d)	No flare
	US	83 F	NR	BNT (2nd)	7 d	Dementia	SubE, eos	NR	NR	TC, SC, DOX, NAM	Ongoing (6 w)	NR
Afacan 2022 [14]	Turkey	88 F	NR	SINV (2nd)	30 d	NR	SubE	DIF: C3/IgG (linear) IIF: NR	NR	TC, SC, MTX	COVID-19 infection while tx	NR
	Turkey	82 F	NR	BNT (3rd)	14 d	NR	SubE	DIF: C3/IgG (linear) IIF: NR	NR	TC, SC, Dapsone	Improved (NR)	NR
	Turkey	65 M	NR	BNT (3rd)	14 d	NR	SubE	DIF: C3/IgG (linear) IIF: NR	NR	TC, DOX	Improved (NR)	NR
	Turkey	82 F	NR	SINV (2nd)	14 d	NR	SubE	DIF: C3/IgG (linear) IIF: NR	NR	TC, SC	Improved (NR)	NR
Agharbi 2022 (1) [5]	Morocco	77 M	Scalp, trunk, and limbs	AZ (1st)	1 d	No past hx	SubE	DIF: IgG (linear) IIF: IgG (linear)	NR	TC, DOX	Improved (NR)	Not received
Alshammari 2022 [29]	Saudi Arabia	78 M	Limbs	BNT (2nd)	1 d	NR	Eos	DIF: C3/IgG/IgM (linear) IIF: NR	NR	TC, SC	Died (2 m)	NR
Avallone 2022 [30]	Italy	72 M	Trunk, lower limbs	MOD (3rd)	20 d	No predisposing factor	SubE, eos	DIF: C3/IgG (linear) IIF: NR	NR	NR	NR	NR
Bailly-Caille 2022 [31]	France	74 M	Limbs	MOD (both)	1st: 10 d 2nd: 2 d	No new meds	SubE, eos	DIF: C3/IgG (linear) IIF: IgG (linear)	BP180−/230−/COL7−/P200+	TC, Colchicine	Resolved (6 m)	NR
Bardazzi 2022 [32]	Italy	76 F	Back, right leg	BNT (3rd)	12 d	NR	NR	NR	BP180+/230+	TC, SC	Resolved (1 m)	NR
	Italy	79 F	Trunk	BNT (3rd)	9 d	NR	NR	NR	BP180+/230+	TC, SC, NAM	Resolved (1 m)	NR
Birabaharan 2022 [33]	US	57 pts	NR	NR	NR	NR	NR	NR	NR	NR	NR	NR
Bostan 2022 [34]	Turkey	67 M	Generalized	Inactivated (1st)	35 d	Under vildagliptin, no past skin hx	SubE, eos	DIF: C3/IgG (linear) IIF: NR	NR	Stop vildagliptin, SC, OMA	Ongoing (8 m)	Flare after both doses
Coto-Segura 2022 [19]	Spain	86 M	Trunk and limbs	BNT (2nd)	17 d	NR	SubE, intraE, eos	DIF: Negative IIF: NR	NR	TC, SC	Resolved (NR)	NR
	Spain	85 M	Trunk and limbs	BNT (2nd)	8 d	NR	SubE, eos	DIF: C3/IgG (linear) IIF: NR	NR	TC, SC	Resolved (NR)	NR
	Spain	84 M	Trunk and limbs	BNT (2nd)	7 d	NR	SubC, eos	DIF: C3/IgG/IgM (linear) IIF: NR	NR	TC, SC	Resolved (NR)	NR
Daines 2022 [35]	US	70s M	Trunk, limbs, palms	BNT (2nd)	1 d	No new meds, DPP4i use	SubE, eos	DIF: C3/IgG (linear) IIF: positive	BP180+/230−	TC, SC, CYSP, MTX	Improved (5 m)	NR
Darrigade 2022 [36]	France	4 pts	NR	NR	NR	NR	NR	NR	NR	NR	NR	NR
Dell’Antonia 2022 [37]	Italy	83 M	1st: legs 2nd: trunk and limbs	BNT (both)	1st: 7 d 2nd: 3 d	No new meds or family hx, DPP4i use	SubE, eos, lym	DIF: C3 (linear) IIF: NR	NR	TC, SC	Resolved (3 w)	Flare after both doses
Desai 2022 [38]	US	73 F	1st: NR 2nd: face, trunk, limbs	MOD (both)	1st: 1 d 2nd: 1 d	No allergic hx, recent illness, or family hx, no new meds	SubE, eos	DIF: C3/IgG (linear) IIF: NR	NR	SC, MMF	Improved (7 d)	Flare after both doses
Fu 2022 [39]	Taiwan	77 M	Trunk and hands	MOD (2nd)	21 d	NR	SubE, neu	DIF: C3/IgG (linear) IIF: negative	NR	SC, CTX	Improved (5 w)	NR
Gambichler 2022 [40]	Germany	80 M	1st: lower legs 2nd: trunk	BNT (both)	1st: 14 d 2nd: NR	No new meds	SubE	DIF: C3/IgG (linear) IIF: IgG (linear)	BP180+/230+	SC	NR	Flare after both doses
	Germany	89 M	Entire integument	BNT (1st)	2 d	No new meds	SubE	DIF: C3/IgG (linear) IIF: IgG (linear)	BP180+/230+	SC	NR	NR
Guo 2022 [13]	China	67 F	Generalized	SINV (1st)	7 d	No new meds, no family hx	SubE, eos	DIF: C3 (linear) IIF: IgG (linear)	BP180+	TC, SC	Improved (2 w)	Flare after both doses
	China	66 F	Generalized	SINV (1st)	10 d	No past hx, no new meds	SubE, eos, neu	DIF: C3 (linear) IIF: IgG (linear)	BP180+	TC, SC	Improved (2 w)	NR
Hali (1) 2022 [41]	Morocco	51 M	Trunk, lower limbs, oral mucosa	AZ (2nd)	7 d	No past hx, no new meds	SubE, eos	DIF: C3 (linear) IIF: IgG (linear)	BP180+	SC	Resolved (4 w)	NR
	Morocco	54 F	Trunk, limbs, oral mucosa	AZ (1st)	3 d	No past hx, no new meds	SubE, eos	DIF: C3/IgG (linear) IIF: C3/IgG (linear)	NR	TC	Improved (NR)	Not received
	Morocco	68 M	1st: vaccination site 2nd: trunk, limbs, oral, genital mucosa	AZ (both)	1st: 14 d 2nd: 7 d	No new meds, no family hx	SubE, eos	DIF: C3 (linear) IIF: NR	NR	SC	Improved (1 m)	NR
Hung 2022 [15]	Taiwan	39 M	Trunk, hands, and feet	MOD (1st)	1 m	NR	SubE, eos	DIF: C3/IgG (linear) IIF: IgG (linear)	NR	SC, DOX	Resolved (NR)	NR
Larson 2021 [42]	US	76 M	Legs	BNT (both)	1st: 21 d 2nd: NR	No new meds, DPP4i use	SubE, eos	DIF: C3/IgG (linear) IIF: IgG (linear)	NR	TC, SC, DOX, NAM	Improved (NR)	Flare after both doses
	US	84 M	Trunk and limbs	MOD (2nd)	14 d	No new/change in meds, DPP4i use	IntraE, eos	DIF: C3/IgG (linear) IIF: NR	NR	TC, SC	Improved (NR)	NR
McMahon 2022 [4]	US	12 pts	Trunk, limbs, oral/genital mucosa	MOD (n = 4) BNT (n = 8)	NR	NR	SubE, eos	DIF: C3/IgG (linear) (n = 5); DIF: IgG (linear) (n = 1) IIF: NR	BP180+ (n = 1)	NR	NR	NR
Maronese 2022 (1) [43]	Italy	84 F	NR	BNT (1st)	25 d	NR	SubE, eos	DIF: C3/IgG (linear) IIF: IgG (linear)	BP180+/230−	TC, SC, DOX	Resolved (3 m)	NR
	Italy	83 M	NR	BNT (1st)	32 d	NR	SubE, eos	DIF: C3/IgG (linear) IIF: IgG (linear)	BP180+/230+	TC, SC, DOX	Resolved (3 m)	NR
	Italy	56 F	NR	MOD (1st)	7 d	NR	SubE, eos	DIF: negative IIF: IgG (linear)	BP180+/230+	TC, DOX	Resolved (3 m)	NR
	Italy	79 M	NR	BNT (1st)	4 d	NR	SubE, eos	DIF: C3/IgG (linear) IIF: IgG (linear)	BP180+/230−	TC, DOX	Resolved (3 m)	NR
	Italy	86 M	NR	BNT (1st)	37 d	NR	SubE, eos	DIF: C3/IgG (linear) IIF: IgG (linear)	BP180+/230−	TC	Resolved (3 m)	NR
	Italy	91 M	NR	BNT (1st)	28 d	NR	SubE, eos	DIF: C3/IgG (linear) IIF: IgG (linear)	BP180−/230−	TC, SC	Resolved (3 m)	NR
	Italy	86 M	NR	BNT (1st)	36 d	NR	SubE, eos	DIF: C3/IgG (linear) IIF: NR	NR	TC, SC, DOX	Resolved (3 m)	NR
	Italy	84 F	NR	MOD (1st)	7 d	NR	SubE, eos	DIF: C3/IgG (linear) IIF: IgG (linear)	BP180+/230−	TC, SC, DOX	Resolved (3 m)	NR
	Italy	84 M	NR	BNT (1st)	23 d	NR	SubE, eos	DIF: C3 (linear) IIF: NR	BP180−/230−	SC	Resolved (3 m)	NR
	Italy	82 F	NR	BNT (1st)	34 d	NR	SubE, eos	DIF: C3/IgG (linear) IIF: NR	BP180−/230−	SC	Improved (3 m)	NR
	Italy	76 M	NR	BNT (1st)	34 d	NR	SubE, eos	DIF: C3 (linear) IIF: NR	BP180−/230−	SC	NR	NR
	Italy	78 M	NR	BNT (1st)	4 d	NR	SubE, eos	DIF: NR IIF: IgG (linear)	BP180+/230+	TC	Resolved (3 m)	NR
	Italy	90 F	NR	BNT (1st)	28 d	NR	SubE, eos	DIF: IgG (linear) IIF: IgG (linear)	BP180+/230−	TC, SC	Improved (3 m)	NR
	Italy	90 M	NR	BNT (1st)	64 d	NR	SubE, eos	DIF: C3 (linear) IIF: negative	BP180−/230−	SC	Resolved (3 m)	NR
	Italy	72 M	NR	BNT (1st)	16 d	NR	SubE, eos	DIF: C3 (linear) IIF: negative	BP180+/230−	TC, SC, MTX	Improved (3 m)	NR
	Italy	80 M	NR	BNT (1st)	6 d	NR	SubE, eos	DIF: C3/IgG (linear) IIF: IgG (linear)	NR	TC, SC	Improved (3 m)	NR
	Italy	77 F	NR	AZ (1st)	3 d	NR	SubE, eos	DIF: C3 (linear) IIF: IgG (linear)	BP180+/230+	MTX	Resolved (3 m)	NR
	Italy	60 F	NR	BNT (1st)	75 d	NR	SubE, eos	DIF: C3 (granular) IIF: IgG (linear)	BP180+/230+	SC	Resolved (3 m)	NR
	Italy	70 F	NR	BNT (1st)	27 d	NR	SubE, eos	DIF: C3 (linear) IIF: IgG (linear)	BP180−/230−	SC	Improved (3 m)	NR
	Italy	72 F	NR	AZ (1st)	7 d	NR	SubE, eos	NR	NR	SC, Dapsone	Improved (3 m)	NR
	Italy	85 M	NR	BNT (1st)	27 d	NR	SubE, eos	NR	NR	SC	Ongoing (3 m)	NR
Maronese 2022 (2) [44]	Italy	85 M	NR	BNT (2nd)	28 d	DPP4i use	SubE, eos	DIF: C3 (linear) IIF: IgG (linear)	BP180+/230+	Stop DPP4i, TC, DOX	Improved (1 m)	NR
	Italy	84 F	NR	BNT (1st)	28 d	DPP4i use for years	NR	NR	BP180−/230−	Stop DPP4i, TC, SC, DOX	Improved (1 m)	NR
	Italy	86 M	NR	BNT (2nd)	14 d	DPP4i use for years	NR	NR	BP180+/230−	Stop DPP4i, TC, SC, DOX	Improved (1 m)	NR
Nakahara 2022 [45]	Japan	71 M	Neck and arms	BNT (2nd)	40 d	DPP4i use for years	SubE, lym	DIF: IgG (linear) IIF: IgG (linear)	BP180+/COL7−	Stop DPP4i, TC, SC, HCQ	Resolved (4 w)	NR
Nida 2022 [46]	US	70 M	Trunk and hands	BNT (2nd)	2 d	New meds of pimavanserin for PD	SubE, eos	DIF: C3/IgG (linear) IIF: NR	NR	TC, SC	Improved (NR)	NR
Pauluzzi 2022 [47]	Italy	46 M	Trunk and upper limbs	BNT (1st)	15 d	No past hx, no new meds	SubE, eos	DIF: C3 (linear) IIF: NR	BP180+	SC, AZA	Improved (7 w)	Not received
Russo 2022 [48]	Italy	75 M	Cutaneous	BNT (1st)	2 d	DPP4i use	NR	NR	NR	Stop DPP4i, TC	Improved (NR)	NR
Savoldy 2022 [49]	US	78 M	1st: back 2nd: trunk, limbs	NR (both)	1st: 7 d 2nd: NR	No new meds, but polypharmacy	SubE, eos	DIF: C3/IgG (linear) IIF: NR	NR	TC, SC, DOX, Dupi	Improved (3 m)	Flare after both doses
Schmidt 2022 [50]	Switzerland	84 F	Both: trunk and limbs	MOD (both)	1st: days 2nd: NR	No new meds, but polypharmacy	SubE, eos	NR	BP180+/230+	NR	NR	Flare after both doses
Shakoei 2022 [51]	Iran	85 F	Trunk and limbs	SINP (1st)	20 d	No allergic, past hx, no new meds	NR	NR	NR	TC, DOX	Improved (NR)	NR
	Iran	91 M	Mucocutaneous	SINP (1st)	19 d	No allergic, past hx, no new meds	NR	NR	NR	TC, RIX	Improved (NR)	NR
Shanshal 2022 [52]	The UK	90 F	Both: trunk, limbs	BNT (both)	1st: 7 d 2nd: NR	No past skin hx, no new meds	SubE, eos	DIF: C3 (linear) IIF: IgG (linear)	NR	1st: TC 2nd: SC	Ongoing (2 m)	Flare after both doses
Wan 2022 [53]	Canada	50 F	3rd: face, neck, trunk, limbs, oral and genital mucosa	BNT (2nd) MOD (3rd)	2nd: 14 d 3rd: 1 d	No new meds	SubE, eos, lym	DIF: C3/IgG (linear) IIF: NR	NR	SC, MTX	Improved (16 w)	NR
	Canada	82 M	Limbs	BNT (both)	1st: 10 d 2nd: 3 d	No new meds	SubE, eos, neu, lym	DIF: C3/IgG (linear) IIF: NR	NR	TC	Resolved (2 w)	Flare after both doses, no flare after the 3rd dose of MOD
Young 2022 [54]	Malta	68 M	Trunk and oral mucosa	BNT (both)	1st: 3 d 2nd: NR	No past hx	SubE, eos	DIF: C3/IgG (linear) IIF: NR	NR	SC, TC	Resolved (3 m)	Flare after both doses
Zhang 2022 [55]	China	23 M	Generalized	SINP (3rd)	1 d	NR	SubE, eos	DIF: C3/IgG (linear) IIF: positive	BP180+/230+	SC	Improved (7 d)	NR
	China	81 M	Limbs and oral mucosa	SINP (3rd)	15 d	NR	SubE	DIF: C3/IgG (linear) IIF: NR	BP180+	SC, IVIG	Improved (NR)	NR
Baffa 2023 [56]	Italy	91 F	Trunk, limbs, and oral mucosa	BNT (2nd)	10 d	No new meds	SubE, eos	DIF: C3/IgG (linear) IIF: IgG (linear)	BP180+	TC, SC, AZA, RIX, Dupi	Resolved (3 m)	NR
Cowan 2023 [57]	Australia	82 M	NR	AZ (2nd)	31 d	NR	NR	NR	NR	NR	NR	NR
	Australia	62 M	NR	BNT (3rd)	123 d	NR	NR	NR	NR	NR	NR	NR
	Australia	71 M	NR	AZ (2nd)	26 d	NR	NR	NR	NR	NR	NR	NR
	Australia	60 F	NR	AZ (2nd)	5 d	NR	NR	NR	NR	NR	NR	NR
Dawoud 2023 [58]	Saudi Arabia	86 M	Generalized	AZ (1st)	1 m	NR	SubE, eos	DIF: C3/IgG (linear) IIF: NR	BP180+/230+	TC, DOX, SC	Improved (7 w)	NR
	Saudi Arabia	76 M	Hands and feet	BNT (1st)	2 wk	NR	SubE, eos	DIF: C3/IgG (linear) IIF: NR	BP180+/230+	TC, DOX, SC	Improved (7 w)	NR
Hsieh 2023 [12]	Taiwan	94 F	Feet, palms, thigh	MOD (1st)	18 d	No new meds	Lym, eos	DIF: C3 (linear) IIF: negative	BP180+	TC, SC, KMnO4	Improved (NR)	NR
Mulianto 2023 [59]	Indonesia	11 M	Generalized	SINV (NR)	4 d	No allergic history or family hx	SubE, eos	DIF: C3/IgG (linear) IIF: NR	NR	SC, ERY	Improved (2 m)	NR
Sun 2023 [60]	Portugal	79 F	Trunk, limbs, mucosa	BNT (2nd)	3 d	No past skin hx, no new meds	SubE, eos, neu	DIF: C3/IgG (linear) IIF: NR	BP180+	TC, SC, IVIG, DOX, MMF	Improved (2 w)	NR
Topal 2023 [61]	Turkey	6 pts (>50 y, 4 F, 2 M)	NR	BNT (2nd) (n = 1) SINV (1st) (n = 2) SINV (2nd) (n = 3)	NR	NR	NR	NR	NR	NR	NR	NR
Üstün 2023 [62]	Turkey	41 F	Trunk, limbs	BNT (1st)	2 wk	No hx of infection or drug use	SubE, eos	DIF: C3/IgG (linear) IIF: NR	NR	TC, SC	Resolved (3.5 m)	NR
Diab 2024 [63]	Iran	70 F	NR	SINP (1st)	20 d	NR	NR	NR	NR	SC	Improved (60 d)	NR
	Iran	77 F	NR	SINP (2nd)	30 d	NR	NR	NR	NR	SC, RIX	Improved (45 d)	NR
Yamamoto 2024 [64]	Japan	72 M	Thigh	BNT (3rd)	1 d	NR	SubE, eos	DIF: C3/IgG IIF: NR	BP180+	SC	Improved (NR)	NR
PGes												
Mustin 2023 [65]	Georgia	36 F	Trunk and limbs	BNT (2nd)	10 d	Pregnancy, no past skin hx	SpD	DIF: C3/IgG (linear) IIF: IgG (linear)	BP180+/BP230−	TC, SC, IVIG	Resolved (7 m)	NR
MMP												
Darrigade 2022 [36]	France	1 pt	NR	NR	NR	NR	NR	NR	NR	NR	NR	NR
Rungraungrayabkul 2023 [66]	Thailand	74 F	Oral mucosa	BNT (1st)	3 wk	No past medical hx, no meds	SubE	DIF: C3/IgG (linear) IIF: NR	NR	TC, DOX	Improved (2 w)	Not received
Calabria 2024 [67]	Italy	72 F	Oral mucosa	BNT (3rd)	9 d	Breast cancer treated with aromatase inhibitor, osteoporosis treated with denosumab	SubE	DIF: IgA/IgG (linear), C3 (granular) IIF: NR	BP180+/BP230−	TC, SC	Resolved (6 w)	NR
LABD												
Coto-Segura 2022 [19]	Spain	71 M	Thighs	BNT (2nd)	3 d	No concomitant meds	SubE, eos	DIF: IgA (linear)	NR	TC	Resolved (NR)	NR
Hali (2) 2022 [68]	Morocco	61 M	Trunk, lower limbs, and oral and genital mucosa	AZ (2nd)	3 d	No infection, no new meds	SubE, eos, lym	DIF: IgA (linear) IIF: IgA (linear)	Dsg1−/3−/BP180−	SC	Improved (NR)	NR
Han 2022 [69]	US	86 F	Neck, trunk, and limbs	MOD (3rd)	1 d	New meds of oral terbinafine for tinea pedis	SubE, neu	DIF: IgA (linear) IIF: NR	NR	TC, SC	Resolved (20 d)	NR
Nahm 2023 [70]	US	66 M	Trunk and limbs	MOD (3rd)	5 d	No new meds	SubE, eos, neu	DIF: IgA/IgM (linear) IIF: IgA	BP180−/230−	TC, SC, Dapsone	Resolved (3 m)	NR
PV												
Solimani 2021 [71]	Asian	40 F	Trunk, back, and oral mucosa	BNT (both)	1st: 5 d 2nd: 3 d	No skin disease hx, no new meds	IntraE, lym, plasma cells	DIF: IgG (IC) IIF: NR	Dsg1+/3+	SC, AZA	Improved (NR)	Flare after both doses
Agharbi 2022 (2) [72]	Morocco	72 F	Head, neck, trunk, limbs, and oral mucosa	BNT (2nd)	7 d	No past hx, no new meds	SupraB, lym	DIF: C3/IgG (IC) IIF: positive	Dsg1+/3+	SC, AZA	Resolved (3 w)	NR
Akoglu 2022 [6]	Turkey	69 F	Mucocutaneous	SINV (2nd)	7 d	No COVID-19 infection/exposure or meds	SupraB	DIF: IgG (IC) IIF: NR	Dsg1+/3+	TC, MTX	Resolved (12 w)	NR
Aryanian 2022 [73]	Iran	43 M	Scalp, face, and oral mucosa	AZ (2nd)	2 d	No past hx, no new meds	NR	NR	NR	SC, AZA	Improved (NR)	NR
Calabria 2022 [74]	Italy	60 F	Oral mucosa	BNT (2nd)	7 d	NR	SupraB, lym, eos	DIF: IgG (IC) IIF: NR	Dsg1−/3+	SC, RIX	Improved (3 w)	NR
Corrá 2022 [75]	Italy	61 F	Face and lower trunk	BNT (3rd)	3 d	No past skin hx	SupraB	DIF: C3/IgG (IC) IIF: IgG (IC)	Dsg1+/3+	SC	NR	NR
	Italy	73 F	Oral mucosa	BNT (3rd)	28 d	No new meds	NR	DIF: C3/IgG (IC) IIF: IgG (IC)	Dsg1−/3+	SC, RIX	NR	NR
	Italy	63 F	Oral mucosa	AZ (both)	1st: 28 d 2nd: 4 d	No past skin hx	IntraE	DIF: C3/IgG (IC) IIF: IgG (IC)	Dsg1+/3+	SC, RIX	Improved (8 w)	Flare after both doses
Das 2022 [76]	India	NR	NR	AZ (2nd)	14 d	NR	NR	NR	NR	NR	NR	NR
Hali (1)2022 [41]	Morocco	58 F	Face, trunk, lower limbs, oral and genital mucosa	BNT (1st)	1 m	NR	IntraE, lym, eos	DIF: C3/IgG (IC) IIF: NR	NR	SC	Improved (NR)	NR
Hatami 2022 [77]	Iran	34 M	Oral mucosa	AZ (NR)	days	No past hx	NR	NR	NR	SC, AZA	NR	NR
Knecht 2022 [78]	Switzerland	89 M	Trunk, left arm, oral mucosa	BNT (2nd)	30 d	Worsened post urology procedure under GA, no past hx	SupraB, lym, his	DIF: IgG (IC) IIF: NR	Dsg1+/3+	SC, RIX	Resolved (10 w)	NR
Koutlas 2022 [79]	US	60 M	Oral mucosa	MOD (2nd)	7 d	No past hx	SupraB	DIF: C3/IgG (IC) IIF: IgG (IC)	Dsg1−/3−	SC, RIX	Resolved (1 m)	NR
Norimatsu 2022 [80]	Japan	86 M	Face, back, upper limbs	BNT (2nd)	1 d	No new meds	SupraB	DIF: IgG (IC) IIF: NR	Dsg1+/3+	TC, SC	Improved (42 d)	NR
Saffarian 2022 [81]	US	76 F	Scalp, upper trunk, oral and genital mucosa	SINP (2nd)	30 d	No past skin hx, no new meds, no DPP4i use	SupraB, eos, lym	DIF: C3/IgG (IC) IIF: NR	Dsg1−/3−	SC, RIX	Improved (NR)	NR
Shakoei 2022 [51]	Iran	30 F	Oral mucosa	SINP (1st)	16 d	No past hx, no new meds	NR	NR	NR	SC, RIX	Improved (NR)	NR
Singh 2022 [82]	India	44 M	Face, neck, trunk, oral mucosa	AZ (2nd)	7 d	No past hx, no new meds	SupraB	NR	Dsg3+	SC, AZA, IVIG	Improved (1 m)	NR
Thongprasom 2022 [83]	Thailand	38 F	Oral mucosa	AZ (1st)	7 d	No allergic hx	NR	NR	NR	TC, steroid mouthwash	Improved (1 w)	NR
Cowan 2023 [57]	Australia	49 F	NR	BNT (3rd)	92 d	NR	NR	NR	NR	NR	NR	NR
Hui 2023 [84]	China	49 F	1st: scalp 2nd: whole body, oral mucosa	SINV (both)	1st: 2 d 2nd: NR	No past hx	IntraE, eos	DIF: IgG (IC) IIF: NR	Dsg1+/3+	SC, AZA, IVIG, MTX, RTX	Improved (8 w)	NR
Khalayli 2023 [85]	Syria	50 F	Limbs, oral and genital mucosa	mRNA (2nd)	10 d	No past hx, no family hx	SupraB	DIF: IgG IIF: NR	NR	TC, SC	Improved (3 w)	NR
Norimatsu 2023 [80]	Japan	86 M	Lumbar region, left arm, face	BNT (2nd)	1 d	Concurrent w/hypopharyngeal and gastric ca	IntraE	DIF: IgG (IC) IIF: NR	Dsg1+/3+	TC, SC	Improved (42 d)	NR
Diab 2024 [63]	Iran	45 M	Oral mucosa	BIV1 (2nd)	20 d	NR	NR	NR	NR	SC, RIX	Improved (60 d)	NR
PF												
Alami 2022 [86]	Morocco	44 M	Face, trunk and limbs	SINP (both)	1st: 7 d 2nd: NR	No past hx, no new meds	IntraE	DIF: IgG (IC) IIF: NR	Dsg1+/3−/ICSA+	SC, AZA	NR	Flare after both doses
Corrá 2022 [75]	Italy	80 M	Face and trunk	BNT (3rd)	17 d	No past skin hx, no new meds	SubC, neu	DIF: Negative IIF: IgG (IC)	Dsg1+	SC, RIX, MMF	NR	NR
	Italy	66 F	Trunk	BNT (2nd)	28 d	No past skin hx	SubC, neu	DIF: IgG (IC) IIF: Negative	Negative	SC, MMF	NR	No flare
Gui 2022 [87]	US	67 F	Trunk	MOD (2nd)	14 d	No past skin hx	IntraE	DIF: C3/IgG (IC) IIF: positive	Dsg1+/3−	TC, SC	Improved (2 m)	NR
Hali (1) 2022 [41]	Morocco	50 F	Scalp and trunk	BNT (2nd)	15 d	No past hx, no new meds	SubC, eos	DIF: C3/IgG (IC) IIF: positive	NR	SC	Resolved (3 w)	NR
Lua 2022 [88]	Singapore	83 M	Scalp, face, trunk, and limbs	BNT (2nd)	2 d	No past skin hx	SpD, eos, plasma cells	DIF: C3 (IC) IIF: IgG (IC)	Dsg1+/3−	SC	Improved (NR)	NR
Pourani 2022 [89]	Iran	75 M	Face and trunk	SINP (3rd)	14 d	No new meds, no hx of COVID-19 pneumonia	IntraE	DIF: C3/IgG (IC) IIF: NR	NR	TC, RIX	Improved (4 w)	NR
Reis 2022 [90]	Caucasian	35 F	Scalp, upper trunk	BNT (2nd)	2 w	No past hx	SubC	DIF: C3/IgG (IC) IIF: positive	Dsg1+/3−	TC, SC	Improved (8 m)	NR
Rouatbi 2022 [91]	Tunisia	70 M	Scalp, trunk, and limbs	BNT (3rd)	7 d	No past skin hx	IntraE	DIF: C3/IgG (IC) IIF: NR	Dsg1+/3−	TC, SC	Improved (3 w)	NR
	Tunisia	48 M	1st: scalp 2nd: face, trunk	AZ (both)	1st: 5 d 2nd: NR	No past hx, no new meds	IntraE	DIF: C3/IgG (IC) IIF: NR	Dsg1+/3−	TC, SC	Resolved (6 m)	Flare after both doses
Yildirici 2022 [92]	Turkey	65 M	1st: scalp, trunk 2nd: neck and trunk	BNT (both)	1st: 30 d 2nd: 14 d	Valsartan-hydrochlorothiazide started 4 m ago	IntraE, neu	DIF: C3/IgG (IC) IIF: NR	Dsg1+/3−	SC, AZA	Improved (2 w)	Flare after both doses
Almasi-Nasrabadi 2023 [93]	The UK	62 F	Face, trunk, and limbs	AZ (both)	1st: 7 d 2nd: 2 d	No past hx, no new meds	SubC, neu	DIF: IgG (IC) IIF: NR	NR	SC, MMF	Improved (NR)	Flare after both doses
Pham 2023 [94]	Vietnam	53 F	Face, trunk, limbs	AZ (4th)	3 w	HTN, no new meds, no family hx	SupraB, lym, neu	DIF: C3/IgG (IC) IIF: NR	NR	SC, RIX	Improved (1 m)	NR
	Vietnam	30 F	Face, neck, trunk	MOD (2nd)	2 m	No family hx	SupraB	DIF: C3/IgG (IC) IIF: NR	NR	TC, SC, TCI	Resolved (4 m)	NR
Weschawalit 2023 [95]	Thailand	NR	NR	AZ (NR)	NR	NR	SubC, neu, eos	DIF: C3/IgG (IC) IIF: NR	NR	NR	NR	NR
Diab 2024 [63]	Iran	30 F	Trunk	SINP (2nd)	14 d	NR	IntraE	NR	NR	RIX	Improved (30 d)	NR
PE												
Falcinelli 2022 [96]	Italy	63 F	Scalp, face, and upper trunk	BNT (2nd)	2 d	NR	SubC	DIF: IgG (IC) IIF: NR	NR	SC	NR	NR
PVeg												
Gui 2022 [87]	Asian	25 M	Face, trunk, limbs, oral and genital mucosa	BNT (2nd)	30 d	No past hx	SupraB, acan	DIF: C3/IgG (IC) IIF: IgG (IC)	Dsg1+/3+	TC, ILOBTX, SC, MMF	Resolved (6 m)	NR
IgA pemphigus												
Lansang 2023 [97]	Canada	64 M	Back, left leg	MOD (NR)	20 d	No new meds	SpD, eos, acantholysis	DIF: C3/IgA/IgG (IC) IIF: NR	NR	TC, IMT	Improved (NR)	NR
Not specified												
Kianfar 2022 [98]	Iran	5 pts	NR	NR (1st) (n = 3) NR (2nd) (n = 2)	NR	NR	NR	NR	NR	NR	NR	NR

acan, acanthosis; AZ, the Oxford-AstraZeneca vaccine; AZA, azathioprine; BIV1, BIV1-CovIran vaccine; BNT, the Pfizer BioNTech (BNT162b2) vaccine; BP, bullous pemphigoid; COL, collagen; CTX, cyclophosphamide; CYSP, cyclosporine; d, day; DIF, direct immunofluorescence; DOX, doxycycline; DPP4i, dipeptidyl peptidase-IV inhibitor; Dsg, desmoglein; Dupi, dupilumab; ELISA, enzyme-linked immunosorbent assay; eos, eosinophils infiltration; ERY, erythromycin; GA, general anesthesia; HCQ, hydroxychloroquine; his, histiocytes infiltration; hx, history; IC, honey-comb-like intercellular pattern; ICSA, anti-intercellular cement substance antibodies; IgG, immunoglobulin G; IIF, indirect immunofluorescence; ILOBTX, intralesional injections of onabotulinum toxin; IntraE, intraepidermal acantholysis; IVIG, intravenous immunoglobulin; IMT, intramuscular triamcinolone; LABD, linear IgA bullous dermatosis; linear, linear pattern along dermo-epidermal junction; lym, lymphocytes infiltration; meds, medications; MMF, mocophenolate mofetil; MMP, mucous membrane pemphigoid; MOD, the mRNA-1273 vaccine; MTX, methotrexate; NAM, nicotinamide; neu, neutrophils infiltration; NR, not recorded; OMA, omalizumab; PD, Parkinson’s disease; PE, pemphigus erythematosus; PF, pemphigus foliaceus; pts, patients; PGes, pemphigoid gestationis; PV, pemphigus vulgaris; PVeg, pemphigus vegetans; SC, systemic corticosteroids; SINP, the Sinopharm BBIBP-CorV vaccine; SINV, the Sinovac CoronaVac vaccine; SpD, spongiotic dermatitis; SubC, subcorneal acantholysis; subE, subepidermal acantholysis; SupraB, suprabasal acantholysis; TC, topical corticosteroids; TCI, topical calcineurin inhibitor; w, week; y, year.

**Table 2 vaccines-12-00465-t002:** Characteristics of the included studies reporting exacerbation of autoimmune bullous dermatosis.

Author, Year	Country	Age, Sex	Blister Sites	Vaccine (Dose)	Onset	Other Triggers	Pathology	DIF/IIF	ELISA	Prior tx	Tx after Flare	Outcome (Time)	Further Vaccine
BP													
Damiani 2021 [18]	Italy	63 F	Trunk	MOD (1st)	1 d	NR	NR	NR	NR	SC	SC	NR	No flare
	Italy	84 M	Widespread, oral mucosa	MOD (both)	14 d	NR	NR	NR	NR	SC, AZA	SC	NR	Flare after both doses
	Italy	82 F	Arms, legs	BNT (1st)	3 d	NR	NR	NR	NR	SC, MMF	SC	NR	No flare
Tomayko 2021 [28]	US	83 M	NR	BNT (1st)	7 d	NR	NR	NR	NR	NR	TC, SC	Ongoing (45 d)	Not received
Afacan 2022 [14]	Turkey	74 F	NR	SINV (1st)	7 d	NR	SubE	DIF: C3/IgG (linear) IIF: NR	NR	NR	TC, SC, DOX, MTX	Improved (NR)	NR
	Turkey	65 F	NR	SINV (2nd)	7 d	NR	SubE	DIF: C3/IgG (linear) IIF: NR	NR	NR	TC, MTX	Improved (NR)	NR
	Turkey	71 M	NR	SINV (2nd)	45 d	NR	SubE	DIF: C3/IgG (linear) IIF: NR	NR	NR	TC, SC, AZA	Improved (NR)	NR
Bardazzi 2022 [32]	Italy	57 F	Trunk, arms	MOD (3rd)	7 d	NR	NR	NR	BP180+/230+	NR	TC, SC, NAM	Resolved (1 m)	NR
	Italy	62 M	Trunk, arms	BNT (3rd)	7 d	NR	NR	NR	BP180+/230+	NR	TC, SC, NAM	Resolved (1 m)	NR
Happaerts 2022 [99]	Caucasian	75 M	Right arm and left buttock	AZ (1st)	10 d	Intake of NSAID once, concomitant AHA, history of COVID-19 pneumonia	NR	NR	NR	SC, NAM, DOX	SC, EMI, rFVII, RIX	Died (15 d)	Not received
Juay 2022 [100]	Singapore	70 F	NR	BNT (1st)	14 d	No new meds, no infection	NR	NR	NR	SC	TC, SC	NR	NR
Martora 2022 [101]	Italy	4 pts (60–80 *, 3M1F)	NR	BNT (2nd) (n = 3) MOD(1st) (n = 1)	5–8 d *	NR	NR	NR	NR	SC+AZA (n = 2) AZA (n = 2)	SC±AZA	Improved (NR)	No flare
Massip 2022 [102]	France	3 pts	NR	NR	1.5–3 d *	NR	NR	NR	NR	NR	NR	NR	NR
Cowan 2023 [57]	Australia	82 M	NR	AZ (NR)	92 d	NR	NR	NR	NR	NR	NR	NR	NR
	Australia	83 M	NR	BNT (NR)	90 d	NR	NR	NR	NR	NR	NR	NR	NR
	Australia	86 F	NR	BNT (NR)	91 d	NR	NR	NR	NR	NR	NR	NR	NR
Rasner 2023 [103]	USA	88 M	Trunk, limbs	BNT (2nd)	1 d	No COVID-19 infection	NR	IIF: IgG	BP180-; BP230+	TC, SC	SC	Improved (5 w)	NR
	USA	69 M	Limbs	MOD (2nd)	14 d	Erythrodermic psoriasis, COVID-19 infection 4 m before	NR	NR	NR	CsA, ADA	TC, ADA	Resolved (6 w)	NR
EBA													
Minakawa 2023 [104]	Japan	20 F	Face, trunk, upper arms, lip	mRNA (1st)	2 d	No medical hx	SubE, neu	DIF: C3/IgG/IgM (linear) IIF: IgG/IgM	BP180-/BP230-/type VII collagen-	SC	SC	Improved (1 w)	NR
PV													
Damiani 2021 [18]	Italy	40 M	Back and upper limbs	MOD (1st)	3 d	NR	NR	NR	NR	RIX	SC, MMF	NR	No flare
	Italy	80 M	Back	BNT (1st)	3 d	NR	NR	NR	NR	SC, MMF	SC	NR	No flare
Akoglu 2022 [6]	Turkey	58 F	Mucocutaneous	SINV (both)	days	No COVID-19 infection/exposure or medical tx	SupraB	DIF: IgG (IC) IIF: NR	Dsg1+/3+	Multiple IMMs	SC, IVIG	Resolved (NR)	Flare after both doses
	Turkey	31 F	Scalp, genital and oral mucosa	BNT (1st)	7 d	No COVID-19 infection/exposure or medical tx	SupraB	DIF: IgG (IC) IIF: NR	Dsg1+/3+	TC	SC	Resolved (8 w)	NR
Avallone 2022 [105]	Italy	46 M	Trunk, arms, oral mucosa	BNT (both)	1st: 5 d 2nd: 5 d	NR	SupraB	DIF: IgG (IC) IIF: NR	Dsg1+/3+	SC, AZA	SC, RIX	Ongoing (NR)	Flare after both doses
Hatami 2022 [77]	Iran	61 M	Scalp and trunk	AZ (NR)	7 d	NR	NR	NR	NR	RIX	SC	NR	NR
Martora 2022 (2) [106]	Italy	7 pts (55–71 *, 4M3F)	NR	BNT (1st) (n = 2) BNT (2nd) (n = 3) MOD (1st) (n = 2)	5–11 d *	NR	NR	NR	NR	SC (n = 1), AZA(n = 6)	SC	NR	NR
Ong 2022 [107]	Asian	46 F	Scalp, trunk, limbs, and oral mucosa	MOD (1st)	7 d	NR	NR	NR	Dsg1+/3+	RIX	SC	Improved (NR)	No flare
Saleh 2022 [108]	Egypt	35 F	NR	SINP (2nd)	5 d	NR	NR	NR	NR	SC	RIX	Improved (NR)	NR
Shakoei 2022 [51]	Iran	28 F	Mucocutaneous	SINP (1st)	14 d	No new meds	NR	NR	NR	SC	SC, RIX	Improved (NR)	NR
Chen 2023 [109]	Taiwan	39 M	Trunk, limbs, oral mucosa	BNT (1st)	7 d	NR	IntraE	DIF: IgG (IC) IIF: NR	NR	TC	SC, RIX, AZA	Improved (NR)	Not received
Cowan 2023 [57]	Australia	32 F	NR	BNT (NR)	6 d	NR	NR	NR	NR	NR	NR	NR	NR
	Australia	73 M	NR	BNT (NR)	15 d	NR	NR	NR	NR	NR	NR	NR	NR
Ligrone 2023 [110]	Italy	56 F	Generalized	MOD (3rd)	5 d	NR	IntraE, supraB	DIF: IgG (IC) IIF: NR	Dsg1+/3+	SC	SC, RIX	Improved (3 w)	NR
PF													
Salmi 2022 [111]	Oman	NR	NR	BNT (NR)	2 d	NR	NR	NR	NR	NR	NR	NR	NR
Rasner 2023 [103]	USA	50 F	NR	BNT (both)	1st: 1 w	NR	NR	IIF: negative	Dsg1+	Not received	TC, SC	Improved (10 w)	NR
Pemphigus													
Massip 2022 [102]	France	2 pts	NR	NR	18 d	NR	NR	NR	NR	NR	NR	NR	NR
Özgen 2022 [112]	Turkey	18 pts	NR	SINV (n = 7) BNT (n = 11)/ 1st (n = 15) 2nd (n = 3)	NR	NR	NR	NR	NR	NR	NR	NR	NR
Not specified													
Kasperkiewicz 2023 [113]	US	84 pts	NR	NR (3rd)	NR	NR	NR	NR	NR	NR	NR	NR	NR
Kianfar 2022 [98]	Iran	66 pts	NR	NR	NR	NR	NR	NR	NR	NR	NR	NR	NR

*, range; AHA, acquired hemophilia A; AZA, azathioprine; BNT, the Pfizer BioNTech (BNT162b2) vaccine; BP, bullous pemphigoid; d, day; DIF, direct immunofluorescence; DOX, doxycycline; DPP4i, dipeptidyl peptidase-IV inhibitor; dsg, desmoglein; EBA, epidermolysis bullosa acquisita; ELISA, enzyme-linked immunosorbent assay; EMI, emicizumab; IC, intercellular pattern; IgG, immunoglobulin G; IIF, indirect immunofluorescence; IMMs, immunomodulators; IVIG, intravenous immunoglobulin; JJ, recombinant adenoviral vector-based Johnson & Johnson vaccine; LABD, linear IgA bullous dermatosis; MMF, mycophenolate mofetil; MOD, the mRNA-1273 vaccine; NAM, nicotinamide; NSAID, nonsteroidal anti-inflammatory drug; PF, pemphigus foliaceus; pts, patients; PV, pemphigus vulgaris; PVeg, pemphigus vegetans; rFVII, recombinant activated factor VII; RIX, rituximab; SC, systemic corticosteroids; SINP, the Sinopharm BBIBP-CorV vaccine; SINV, the Sinovac CoronaVac vaccine; SupraB, suprabasal acantholysis; Tx, treatment; w, week; y, year.

**Table 3 vaccines-12-00465-t003:** Summary of characteristics of the included studies.

AIBD Type	Study (n)	Patient (n)	Country	Age *	Sex	Vaccine	Dose	Onset *	Outcome	Time to Improvement/Resolution *	Further Vaccine
New onset											
BP	47	174	Asia 30 (17.24%) Africa 4 (2.30%) America 90 (51.72%) Europe 46 (26.44%) Oceania 4 (2.30%)	11–97	M 58 (57.43%) F 43 (42.57%) NR 73	AZ 10 (8.93%) MOD 19 (16.96%) SINV 10 (8.93%) SINP 6 (5.36%) Inactivated 1 (0.89%) BNT 66 (58.93%) NR 62	1st 44 (44.00%) 2nd 32 (32.00%) 3rd 9 (9.00%) Both 15 (15.00%) NR 74	1 d–123 d	Died 1 (1.18%) Improved 44 (51.76%) Resolved 31 (36.47%) Ongoing 8 (9.41%) Other 1 (1.18%) NR 89	1 w–6 m	No flare (2nd) 2 (9.52%) Flare (both) 14 (66.67%) Not received 5 (23.81%) NR 153
PGes	1	1	Europe 1 (100.00%)	36	F 1 (100.00%)	BNT 1 (100.00%)	2nd 1 (100.00%)	10 d	Resolved 1 (100.00%)	7 m	NR 1
MMP	3	3	Asia 1 (33.33%) Europe 2 (66.67%)	72–74	F 2 (100.00%) NR 1	BNT 2 (100.00%) NR 1	1st 1 (50.00%) 3rd 1 (50.00%) NR 1	9 d–3 w	Improved 1 (50.00%) Resolved 1 (50.00%) NR 1	2 w–6 w	Not received 1 (100.00%) NR 2
LABD	4	4	Africa 1 (25.00%) America 2 (50.00%) Europe 1 (25.00%)	61–86	M 3 (75.00%) F 1 (25.00%)	AZ 1 (25.00%) MOD 2 (50.00%) BNT 1 (25.00%)	2nd 2 (50.00%) 3rd 2 (50.00%)	1 d–5 d	Improved 1 (75.00%) Resolved 3 (25.00%)	20 d–3 m	NR 4
PV	21	23	Asia 13 (56.52%) Africa 2 (8.70%) America 2 (8.70%) Europe 5 (21.74%) Oceania 1 (4.35%)	30–89	M 8 (36.36%) F 14 (63.64%) NR 1	AZ 6 (26.09%) MOD 1 (4.35%) SINV 2 (8.70%) SINP 2 (8.70%) BNT 10 (43.48%) BIV1 1 (4.35%) mRNA 1 (4.35%)	1st 3 (13.64%) 2nd 13 (59.09%) 3rd 3 (13.64%) Both 3 (13.64%) NR 1	1 d–92 d	Improved 14 (77.78%) Resolved 4 (22.22%) NR 5	1 w–12 w	Flare (both) 2 (100.00%) NR 21
PF	13	16	Asia 7 (43.75%) Africa 4 (25.00%) America 1 (6.25%) Europe 4 (25.00%)	30–83	M 7 (46.67%) F 8 (53.33%) NR 1	AZ 4 (25.00%) MOD 2 (12.5%) SINP 3 (18.75%) BNT 7 (43.75%)	2nd 7 (46.67%) 3rd 3 (20.00%) 4th 1 (6.67%) Both 4 (26.67%) NR 1	2 d–2 m	Improved 9 (75.00%) Resolved 3 (25.00%) NR 4	2 w–8 m	No flare (2nd) 1 (20.00%) Flare (both) 4 (80.00%) NR 11
PE	1	1	Europe 1 (100%)	63	F 1 (100.00%)	BNT 1 (100.00%)	2nd 1 (100.00%)	2 d	NR 1	NR	NR 1
PVeg	1	1	Asia 1 (100%)	25	M 1 (100.00%)	BNT 1 (100.00%)	2nd 1 (100.00%)	30 d	Resolved 1 (100.00%)	6 m	NR 1
IgA pemphigus	1	1	America 1 (100.00%)	64	M 1 (100.00%)	MOD 1 (100.00%)	NR 1	20 d	Improved 1 (100.00%)	NR	NR 1
Not specified	1	5	Asia 5 (100.00%)	NR	NR 5	NR 5	1st 3 (60.00%) 2nd 2 (40.00%)	NR	NR 5	NR	NR 5
Total	83	229	Asia 57 (24.89%) Africa 11 (4.80%) America 96 (41.92%) Europe 60 (26.20%) Oceania 5 (2.18%)	11–97	M 78 (52.70%) F 70 (47.30%) NR 81	AZ 21 (13.04%) MOD 25 (15.53%) SINV 12 (7.45%) SINP 11 (6.83%) Inactivated 1 (0.62%) BNT 89 (55.28%) BIV1 1 (0.62%) mRNA 1 (0.62%) NR 68	1st 51 (33.77%) 2nd 59 (39.07%) 3rd 18 (11.92%) 4th 1 (0.66%) Both 22 (14.57%) NR 78	1 d–123 d	Died 1 (0.81%) Improved 70 (56.45%) Ongoing 8 (6.45%) Other 1 (0.81%) Resolved 44 (35.48%) NR 105	1 w–8 m	No flare (2nd) 3 (10.34%) Flare (both) 20 (68.97%) Not received 6 (20.69%) NR 200
Flare											
BP	10	23	Asia 4 (17.39%) America 3 (13.04%%) Europe 13 (56.52%) Oceania 3 (13.04%)	57–88	M 12 (60.00%) F 8 (40.00%) NR 3	AZ 2 (10.00%) MOD 5 (25.00%) SINV 3 (15.00%) BNT 10 (50.00%) NR 3	1st 7 (41.18%) 2nd 7 (41.18%) 3rd 2 (11.76%) Both 1 (5.88%) NR 6	1 d–92 d	Died 1 (7.69%) Improved 8 (61.54%) Ongoing 1 (7.69%) Resolved 3 (23.08%) NR 10	1 m–45 d	No flare (2nd) 6 (66.67%) Flare (both) 1 (11.11%) Not received 2 (22.22%) NR 14
EBA	1	1	Asia 1 (100.00%)	20	F 1 (100.00%)	mRNA 1 (100.00%)	1st 1 (100.00%)	2 d	Improved 1 (100.00%)	1 w	NR 1
PV	12	20	Asia 6 (30.00%) Africa 1 (5.00%) Europe 11 (55.00%) Oceania 2 (10.00%)	28–80	M 10 (50.00%) F 10 (50.00%)	AZ 1 (5.00%) MOD 5 (25.00%) SINV 1 (5.00%) SINP 2 (10.00%) BNT 11 (55.00%)	1st 10 (58.82%) 2nd 4 (23.53%) 3rd 1 (5.88%) Both 2 (11.76%) NR 3	3 d–15 d	Improved 5 (62.50%) Ongoing 1 (12.50%) Resolved 2 (25.00%) NR 12	3 w–8 w	No flare (2nd) 3 (50.00%) Flare (both) 2 (33.33%) Not received 1 (16.67%) NR 14
PF	2	2	Asia 1 (50.00%) America 1 (50.00%)	50	F 1 (100.00%) NR 1	BNT 2 (100.00%)	Both 1 (100.00%) NR 1	2 d–1 w	Improved 1 (100.00%) NR 1	10 w	NR 2
Pemphigus	2	20	Asia 18 (90.00%) Europe 2 (10.00%)	NR	NR 20	SINV 7 (38.89%) BNT 11 (61.11%) NR 2	1st 15 (83.33%) 2nd 3 (16.67%) NR 2	18 d	NR 20	NR	NR 20
Not specified	2	150	Asia 66 (44.00%) America 84 (56.00%)	NR	NR 150	NR 150	3rd 84 (100.00%) NR 66	NR	NR 150	NR	NR 150
Total	24	216	Asia 96 (44.44%) Africa 1 (0.46%) America 88 (40.74%) Europe 26 (12.04%) Oceania 5 (2.31%)	20–88	M 22 (52.38%) F 20 (47.62%) NR 174	AZ 3 (4.92%) MOD 10 (16.39%) SINV 11 (18.03%) SINP 2 (3.28%) BNT 34 (55.74%) mRNA 1 (1.64%) NR 155	1st 33 (23.91%) 2nd 14 (10.14%) 3rd 87 (63.04%) Both 4 (2.90%) NR 78	1 d–92 d	Died 1 (4.35%) Improved 15 (65.22%) Ongoing 2 (8.70%) Resolved 5 (21.74%) NR 193	1 w–10 w	No flare (2nd) 9 (60.00%) Flare (both) 3 (20.00%) Not received 3 (20.00%) NR 201

*, range; AIBD, autoimmune bullous dermatosis; BP, bullous pemphigoid; LABD, linear IgA bullous dermatosis; PV, pemphigus vulgaris; PF, pemphigus foliaceus; PGes, pemphigoid gestationis; PVeg, pemphigus vegetans; DIF, direct immunofluorescence; Ab, antibody; d, day; w, week; y, year.

## Data Availability

No new data were generated in support of this research.

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
