# Peer review of "New Onset and Exacerbation of Autoimmune Bullous Dermatosis Following COVID-19 Vaccination: A Systematic Review"

_vaccines, 2024, doi:10.3390/vaccines12050465_

Round 1
Reviewer 1 Report
Comments and Suggestions for Authors
In this manuscript, authors performed a systematic review to evaluate autoimmune bullous dermatoses following COVID-19 vaccination. The topic of the manuscript is important and clinically relevant, the methodology is overall thorough. However, there are several issues to be addressed by the authors:
Major issues:
1. It is not clear from the title that they not only assessed the development of AIBD but also flares
2. “mRNA vaccines were commonly associated with the development of AIBD.” this statement should be revised
3. Given that the topic is very much current, it is questionable that the search was conducted only until 16th January 2023, it should be updated
4. Why were further AIBDs not included in the search key?
5. Conclusions should be more cautious as we do not know if the reported AIBDs would have developed without vaccination as well
Minor issue:
1. In the keywords, instead of “systematic review”, further AIBDs should be listed
Author Response
Many thanks for your comments. Please see the uploaded 'Reply to Reviewer 1' WORD file.

Reviewer 2 Report
Comments and Suggestions for Authors
Dear AA
Given the infrequency and common misdiagnosis of autoimmune bullous diseases (AIBDs), I find the subject matter of your manuscript both timely and relevant, meriting publication. We have provided specific comments aimed at enhancing your text to make it more accessible and comprehensible to readers. It is particularly important, as suggested, to clarify certain aspects within the discussion section to underscore these points.
Line 35
We suggest the AA to mention in the bibliography ( DOI: 10.1111/dth.15153 Cutaneous adverse reactions after COVID-19 vaccines in a cohort of 2740 Italian subjects: An observational study) published in 2021.
Lines 85-86
We recommend the authors to precisely detail if the Elisa test has been applied for the detection of pemphigus-related auto-antibodies (Desmoglein 1 and Desmoglein 3) in the serum of patients. It is interesting to clearly report the presence or absence of these auto-antibodies in both new-onset and exacerbated cases of autoimmune bullous diseases (AIBDs) within the study cohort.
Lines 205-208
Dear AA
Considering the chronic relapsing nature of autoimmune bullous diseases (AIBDs), it is not definitively established whether AIBDs, post-COVID-19 vaccination and successfully treated, have reached complete remission or the patients mentioned in this paragraph needed further monitoring. We suggest the authors to explicitly specify this point in the reported paragraph later analysed in the discussion chapter.
Author Response
Many thanks for your comments. Please see our uploaded WORD file 'Reply to Reviewer 2 letter 20240414'.

Reviewer 3 Report
Comments and Suggestions for Authors
Thank you for the opportunity to review this interesting manuscript. Drug induced AIBDs constitute established though uncommon clinical diagnoses and vaccines have been reported as triggers of disease in genetically susceptible patients. This paper is pleasant to read and well structured; moreover, the tables are informative of the cases retrieved from the literature.
I suggest the authors to consider the following comments:
- Lines 38-41: in your introduction it would be useful to distinguish between disorders characterized by intraepidermal detachment (pemphigus group) and the subepidermal diseases (or pemphigoid group)
- Line 141: most patients with disease flare had a diagnosis of pemphigus. Considering that pemphigus is less common that pemphigoid, what could be a possible explanation? Maybe the more chronic nature of the disorder?
- most reactions are reported after mRNA vaccines, however you should emphasize that these were also the most widely employed;
- Lines 153-154 I suggest clarifying these percentages to avoid a potential source of confusion. What do you mean by “most cases” since 22 + 22% is less than 50%?
- Lines 193-194 and line 205: same as above
- considering the reported timing between dose and manifestation is variable and often very short, what could be the possible pathogenic explanations of such timing? (also consider the point below)
- line 261-262 “The etiology and pathogenesis of AIBD remain largely elusive”. I suggest mentioning in the discussion the role of T cells in these disorders. Since autoreactive T cells may be found in both AIBD patients and healthy subjects, could the vaccine promote an “imbalance between Th2 responses against cutaneous antigens, which are associated with disease, and Th1 regulatory responses, which maintain immune tolerance and are dysfunctional in disease” (Hertl, M.; Eming, R.; Veldman, C. T Cell Control in Autoimmune Bullous Skin Disorders. J. Clin. Investig. 2006, 116, 1159–1166)
- could it be useful to investigate alleles that may be linked with a genetic predisposition with (drug-induced)-AIBD? Consider reading: Sernicola, A.; Mazzetto, R.; Tartaglia, J.; Ciolfi, C.; Miceli, P.; Alaibac, M. Role of Human Leukocyte Antigen Class II in Antibody-Mediated Skin Disorders. Medicina2023,59,1950.
Comments on the Quality of English Language- Minor writing mistakes, for example line 36 “has” should be “have” and line 210 “systemic”
Author Response
Many thanks for your comments. Please see our uploaded WORD file 'Reply to Reviewer 3 letter 20240414'.
